# Evaluation of Triggered Electromyogram Monitoring during Insertion of Percutaneous Pedicle Screws

**DOI:** 10.3390/jcm11051197

**Published:** 2022-02-23

**Authors:** Hayato Futakawa, Shigeharu Nogami, Shoji Seki, Yoshiharu Kawaguchi, Masato Nakano

**Affiliations:** 1Department of Orthopedic Surgery, Takaoka City Hospital, Toyama 933-8550, Japan; smacnog@mac.com (S.N.); mnakano-tym@umin.ac.jp (M.N.); 2Department of Orthopedic Surgery, Faculty of Medicine, University of Toyama, Toyama 930-0194, Japan; seki@med.u-toyama.ac.jp (S.S.); zenji@med.u-toyama.ac.jp (Y.K.)

**Keywords:** percutaneous pedicle screw, minimally invasive surgery, minimally invasive spine stabilization, triggered electromyogram monitoring, spine surgery, complication

## Abstract

Objective: percutaneous pedicle screw (PPS) fixation has been widely used in minimally invasive spine stabilization. Triggered electromyogram (TrEMG) monitoring is performed to prevent PPS misplacement, but is not widely accepted. We have newly developed an insulating tap device to minimize the misplacement of PPS. Methods: TrEMG was measurable in insulation tap devices in 31 cases, and in non-insulating tap devices in 27 cases. Fluoroscopy was used to insert 194 PPS and 154 PPS, respectively. Based on the Rampersaud classification of postoperative computed tomography, we classified PPS insertion into four categories (Grade A as no violation, Grade D as more than 4 mm perforation). Results: Grade A was noted in 168 PPSs (86.6%) and Grade B to D in 26 PPSs in the insulation tap device group, and Grade A was noted in 129 PPSs (83.8%) and Grade B to D in 25 PPSs in the non-insulating tap device group, respectively. At a cutoff value of 11 mA, the sensitivity was 41.4% and the specificity was 98.2%. The sensitivity and specificity of the non-insulating tap device were 4.0% and 99.2%, respectively. Conclusions: The insulation treatment of the tap device has improved the sensitivity of TrEMG. TrEMG using the insulating tap device is one of the methods for safe PPS insertion.

## 1. Introduction

The fixation method using the percutaneous pedicle screw (PPS) is essential to minimally invasive spine stabilization (MISt) surgery and has been widely used in recent years. However, the placement of a PPS is technically demanding, and misplacement has been reported in 4.7% to 23% of cases [1,2,3,4,5]. To prevent misplacement of PPS, there are several previous papers using navigation and various insertion devises [6,7]. These devices improved the accuracy of PPS placement, but they needed expensive hardware and not all medical institutions could introduce these devices. The reliability of triggered electromyogram (TrEMG) monitoring during open pedicle screw placement has been established [8], but the reliability of TrEMG monitoring during PPS placement is controversial [9,10]. Accurate EMG measurement requires adequate insulation of the stimulation device in addition to a dry surgical field. Therefore, we developed the insulated coated tap device and evaluated the accuracy of PPS placement using our device as stimulators for TrEMG monitoring, and compared the result with using a non-insulation tap.

## 2. Materials and Methods

Retrospective study was carried out. We reviewed data of 31 patients who underwent spine surgery with PPS and measured TrEMG using the insulation tap device at our institution between August 2015 and February 2017 (Figure 1). This result was compared with that of 27 patients measuring TrEMG using a non-insulation tap between April 2014 and August 2015 (Table 1). We used fluoroscopy-inserting 194 PPS in the insulation tap group and 154 PPS in the non-insulation tap group. We inserted PPS by standardized method (inserting guidewire into the vertebral body through pedicle using fluoroscopy, tapping, inserting PPS). When we confirmed the tip of the insulation tap passing beyond the posterior wall of the vertebral body by fluoroscopy, we measured TrEMG connecting clips to this tap device and measuring the threshold of muscle action potential (Figure 2). NVM5^®^ Nerve Monitoring System (NuVasive, San Diego, CA, USA) was used as a measuring assembly. This system displayed threshold as a numerical value, and gave beep sound: 11 mA or more, 7 to 10 mA, and 6 mA or less [11]. Based on the Rampersaud classification of postoperative computed tomography, we classified PPS insertion into four categories, Grade A as no violation, Grade B as less than 2 mm perforation, Grade C as 2–4 mm perforation, and Grade D as more than 4 mm perforation [12]. When this system displayed 10 mA or less, we rechecked the C-arm image and if we could confirm malposition of the guide wire, we re-inserted it. Two screws that were misplaced at the time of tapping and replaced in the correct position were classified as Grade D. We calculated sensitivity and specificity of TrEMG monitoring from these results. This study had been approved by the ethical committee in our hospital and we obtained informed consent from the patients before operations.

## 3. Results

The level of vertebrae in the insulation tap group was Th10 (2), Th11 (4), Th12 (6), L1 (4), L2 (31), L3 (38), L4 (50), L5 (47), and S1 (12), and that in the non-insulation tap group was Th9 (2), Th10 (2), Th11 (6), Th12 (4), L1 (8), L2 (18), L3 (22), L4 (38), L5 (38), and S1 (16), respectively. Of 194 PPS using the insulating tap device, 179 were 11 mA or more, 13 were 7 to 10 mA, 2 were 6 mA or less, and of 154 using the non-insulating tap device, 152 were 11 mA or more, 2 were 7 to 10 mA. The minimum current value was 6 mA. On the postoperative CT, Grade A was noted in 168 PPSs, and Grades B–D that had pedicle perforations due to PPS in 26 PPSs (13.4%) in the insulating tap group and Grade A was noted in 129 PPSs, and Grades B–D in 25 PPSs (16.1%) in the non-insulating tap group, respectively (Table 2). The level of vertebrae perforating the medial wall of the pedicle was Th12 (2), L2 (4), L3 (4), L4 (9), L5 (6), and S1 (1) in the insulating tap group, and Th9 (1), Th10 (1), Th11 (1), Th12 (1), L2 (2), L3 (2), L4 (9), L5 (6), and S1 (2) in the non-insulating tap group, respectively (Table 3). All misplacement cases were asymptomatic. At a cutoff value of 11 mA, the sensitivity was 35.0% and the specificity was 96.4% in the insulating tap group, and the sensitivity was 4% and the specificity was 99.2% in the non-insulating tap group, respectively (Table 4). When we consider pedicle fracture as perforation of pedicle (Grades B–D), the sensitivity was 41.4% and the specificity was 98.2% (Figure 3 and Table 5). The insulation treatment for a screw tap improved the accuracy of inserting PPS statistically (*p* < 0.05: Chi-squared test).

## 4. Discussion

The utility of TrEMG monitoring on the placement of pedicle screw had been reported previously [8,11], but these studies stimulated pedicle screws directly, and there were few affirmative reports about that of PPS. The utility of using dynamic EMG and intraoperative fluoroscopy together for secure and accurate placement of PPS had been reported previously [11,12]. In the study, the perforation rate of pedicle had been about 15%, which was the equivalent rate as compared with our rate in the non-insulating tap group. On the other hand, it had been reported the inaccuracy of the free-running EMG and TrEMG using a Jamshidi needle [10,13]. In the series of procedures of inserting PPS (inserting guide wire, tap, screw), we expected that it was most appropriate to measure the TrEMG inserting tap, because if we could insert the guide wire in the correct position in pedicles, we might the perforate pedicle wall on tapping. Tap stimulation is better electrical conductivity than direct stimulation to the PPS. Furthermore, stimulators must not contact with soft tissue electrically for accurate electrical stimulations. The insulating “sleeve” was used in this study, but the results were 4% sensitivity and 99.2% specificity in the non-insulating group. That was because the insulating sleeve could not completely prevent electric leakage through bleeding, so we insulated the shaft of the tap device, and used it as a stimulator. This process improved the sensitivity from 4% to 35%. Moreover, we found out 3 cases of pedicle fracture or lateral breach of PPS postoperative CT scan, and all these 3 cases showed 10 mA or less on TrEMG monitoring (Figure 3). This proved that pedicle fracture or lateral breach of PPS reacted TrEMG monitoring. In this study, we adopted a cutoff value of 11 mA because of the setting of the Nerve Monitoring System. If we adopted a conventional cutoff value 6 mA, the sensitivity was 1% and 92.3% specificity in the insulating tap group. So, we thought this cutoff value was partially credible. On the other hand, we thought the sensitivity of this method was not enough. The insulating tap of this study could not completely prevent leakage of electrical stimulation, so we need to develop a better insulating tap.

There are some limitations in this study. At first, this study was a retrospective consequent study. Secondly, this study was carried out in a single institution by a single surgeon (M.N.), although another co-author (S.N.) evaluated blindly postoperative CT grade. Thirdly, the operations of the insulating tap group were done in later years than the non-insulating tap group, so there might be vias of technical proficiency of the surgeon. Lastly, there was no control group in this study design.

## 5. Conclusions

TrEMG monitoring using insulation treatment of a tap device has improved the sensitivity of TrEMG. TrEMG using an insulating tap device is one of the methods for safe PPS insertion.

## Figures and Tables

**Figure 1 jcm-11-01197-f001:**
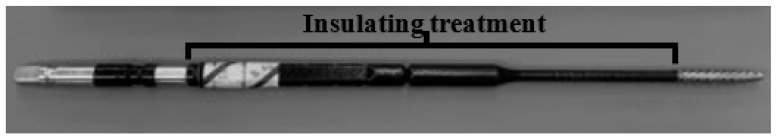
Newly developed insulated coated tap device was used in this study.

**Figure 2 jcm-11-01197-f002:**
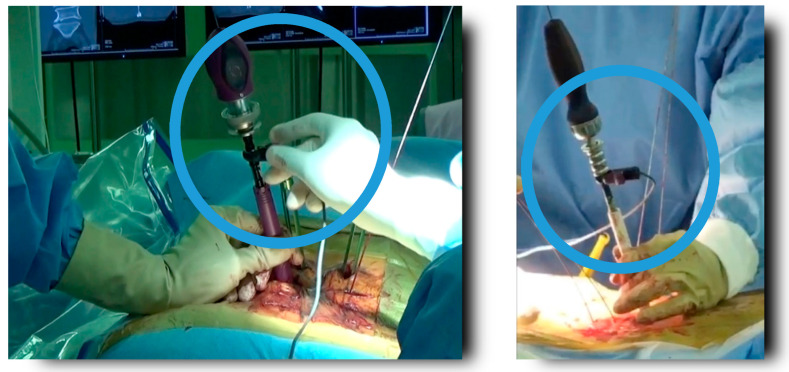
Before inserting the percutaneous pedicle screw, a TrEMG connecting clip was connected to the non-insulated part of the insulating tap and TrEMG was measured (NVM5^®^ V2.0 Nerve Monitoring System: NuVasive, San Diego, CA, USA).

**Figure 3 jcm-11-01197-f003:**
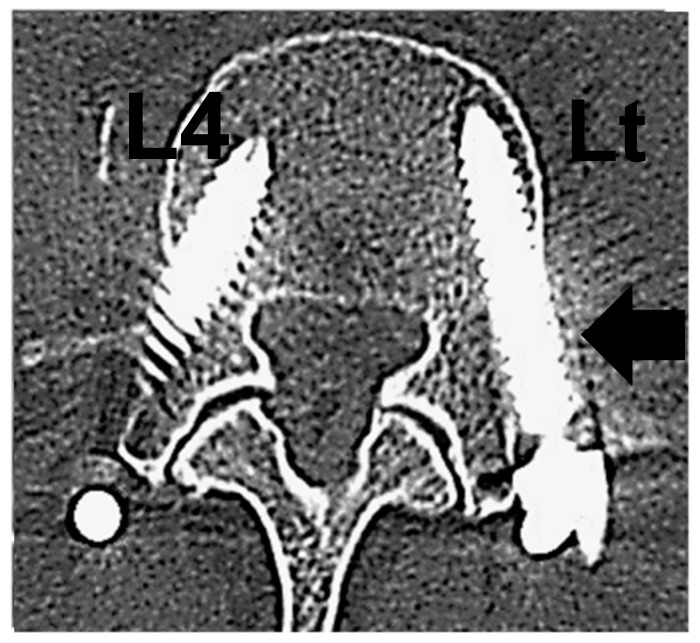
This is a case of L3-4 lumbar lateral interbody fusion and percutaneous pedicle screw for lumbar spinal stenosis. Left L4 monitor indicated 8 mA. Postoperative CT scan showed lateral perforation of the left L4 pedicle on the tapping and pedicle fracture.

**Table 1 jcm-11-01197-t001:** Comparison of the baseline data between two groups.

Item	Insulating Tap Group (*n* = 31)	Non-Insulating Tap Group (*n* = 27)
Sex (M:F, *n*)	12:19	12:15
Age (mean ± SD, years)	68.7 ± 12.4	67.3 ± 13.0
Number of levels (mean ± SD, min)	3.2 ± 1.1	2.9 ± 1.0
Indication for spine surgery (*n*)		
Degenerative spondylosis	30	27
Trauma	1	0

**Table 2 jcm-11-01197-t002:** The result of TrEMG and postoperative computed tomography.

Insulating Tap Group					
	Grade A	Grade B	Grade C	Grade D	
Green ≥ 11 mA	160	18	1	0	179
Yellow 7–10 mA	8	5	0	0	13
Red ≤ 6 mA	0	0	0	2	2
	168	23	1	2	194
Non-Insulating Tap Group					
	Grade A	Grade B	Grade C	Grade D	total
Green ≥ 11 mA	128	22	1	1	152
Yellow 7–10 mA	1	1	0	0	2
Red ≤ 6 mA	0	0	0	0	0
total	129	23	1	1	154

**Table 3 jcm-11-01197-t003:** The level of vertebrae perforating the medial wall of the pedicle in each group.

	Insulating Tap	Non-Insulating Tap
Th9		1 *(2)
Th10	0 *(2)	1 *(2)
Th11	0 *(4)	1 *(6)
Th12	2 *(6)	1 *(4)
L1	0 *(4)	0 *(8)
L2	4 *(31)	2 *(18)
L3	4 *(38)	2 *(22)
L4	9 *(50)	9 *(38)
L5	6 *(47)	6 *(38)
S	1 *(12)	2 *(16)
total	26 *(194)	25 *(154)

* (Total number of inserting PPS).

**Table 4 jcm-11-01197-t004:** The sensitivity and specificity of TrEMG.

Insulating Tap Group		
Threshold	Pedicle breach +	Pedicle breach −
<11 mA	True Positive 9	False Positive 6
≥11 mA	False Negative 17	True Negative 162
	*Sensitivity 35.0%	*Specificity 96.4%
Non-Insulating Tap Group		
Threshold	Pedicle breach +	Pedicle breach −
<11 mA	True Positive 1	False Positive 1
≥11 mA	False Negative 24	True Negative 128
	*Sensitivity 4.0%	*Specificity 99.2%

* The accuracy at a cutoff value 11 mA.

**Table 5 jcm-11-01197-t005:** The sensitivity and specificity of TrEMG including 3 cases of pedicle fracture on postoperative computed tomography counted as pedicle breach.

Insulating Tap Group		
Threshold	Pedicle breach +	Pedicle breach −
<11 mA	True Positive 12	False Positive 3
11 mA≤	False Negative 17	True Negative 162
	*Sensitivity 41.4%	*Specificity 98.2%

* The accuracy at a cutoff value 11 mA.

## Data Availability

The data that support the findings of this study are available from the corresponding author upon reasonable request.

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
