# Peer review of "Evaluation of Triggered Electromyogram Monitoring during Insertion of Percutaneous Pedicle Screws"

_jcm, 2022, doi:10.3390/jcm11051197_

Round 1

Reviewer 1 Report

Futakawa et al. studied a new developed insulated tap device in percutaneous pedicle screw fixation, which improved the sensitivity compared to the non insulation device group. While the manuscript has some novelty, some comments should be adressed, in particular more details in the methods.

Major comments:
- The manuscript has multiple sections that require English language corrections, which in its present form is insufficient.
- The method section requires more elaboration on the techniques used.|
Could you please elaborate more on the technique for percutaneous spinal instrumentation? Likewise, more in depth detail of the triggered EMG technique would be of added value to the manuscript.
- Could you please provide more details on the patient characteristics of the population? E.g Age, sex, the number of levels in each patient? What were the indications for spine surgery. e.g. deformative or traumatic
- Were there no lateral breaches in the patient population? If there were lateral breaches, the analysis in the current manuscript is inaccurate as those breaches would have to be classified as True Negatives, consistent with the literature:  PMID 18007243
- The authors state that there is a statistical significant difference between sensitivities in the groups. Could the authors please add a p-value, and specify which statistical test was used?

Minor comments:
- Please consequently use abbreviations: the introduction uses both "Trigger EMG" and "TrEMG", which should only be 1 of the two
- There is a descrepancy in the results section between the medial breaches in non-insulating tap group in the text and in the table: S1 has 2 breaches in the text, which is a 1 in the table
- Please delete the last sentence of the conclusion as it does not have additional merit to the conclusion
-In the abstract please also include the sensitivity and specificity of the non insulating group, as it is difficult to assess the improvement in sensitivity without this in the abstract.

Reviewer 2 Report

I commend the efforts of the authors to find innovative ways to improve the sensitivity of detection of the pedicle breach to improve the safety of the PPS insertion. However, the manuscript presented could be improved in the following domains.

  1. A structured abstract can be presented
  2. Introduction can be elaborated in terms of other methods available in the market to improve the safety of the PPS insertion
  3. The general information of the two groups compared could be presented in a single table rather than many split tables for each
  4. The discussion could also be made in line of the conventional cut-off used for detection of the breach and the improvement in the current cut-off derived from the study
  5. Discussion could also be made on the other improvement that is worth investigation to improve the precision and safety of PPS insertion 
  6. Lot of language errors that needs revision with the help of a native language expert. 

Reviewer 3 Report

Well written manuscript with novelty 

Accept with minor revision 

Author Response

We revised some points according to reviewer's advice.

Thank you for your comment.